# Proteomic Approach to *Anemonia sulcata* and Its Symbiont *Symbiodinium* spp. as New Source of Potential Biotechnological Applications and Climate Change Biomarkers

**DOI:** 10.3390/ijms241612798

**Published:** 2023-08-14

**Authors:** Ana Maria Melendez-Perez, Almudena Escobar Niño, Rafael Carrasco-Reinado, Laura Martin Diaz, Francisco Javier Fernandez-Acero

**Affiliations:** 1Physical Chemical Department, Institute of Marine Research (INMAR), International Campus of Excellence of the Sea (CEIMAR), Faculty of Marine and Environmental Sciences, University of Cadiz, 11510 Puerto Real, Spain; ammelendez2@gmail.com; 2Center for Research and Innovation in Biodiversity and Climate Change (ADAPTIA), Faculty of Engineering, University Simón Bolívar, Barranquilla 59-65, Colombia; 3Microbiology Laboratory, Institute for Viticulture and Agri-Food Research (IVAGRO), Faculty of Environmental and Marine Sciences, Department of Biomedicine, Biotechnology and Public Health, University of Cádiz, 11510 Puerto Real, Spain; almudena.escobar@uca.es (A.E.N.); rafael.carrasco@uca.es (R.C.-R.)

**Keywords:** climate change, proteomics, *Anemonia sulcate*, *Symbiodinium* spp., biomarkers

## Abstract

Marine ecosystems are among the richest in terms of biodiversity, and at present, still remain largely unknown today. In the molecular biology era, several analyses have been conducted to unravel the biological processes in this ecosystem. These systems have provided biotechnological solutions to current problems, including the treatment of diseases, as well as for the development of new biotechnological tools with applications in biomedicine and/or agri-food. In addition, in the context of climate change and global warming, these studies become even more necessary for the development of molecular tools that allow a reliable follow-up of this situation to anticipate alterations and responses of bioindicator species and to create a database to prevent and predict the environmental and climatic changes before the damage is irreversible. Proteomics approaches have revealed their potential use to obtain the set of biological effectors that lead to the real biological station on a specific stage, the proteins. In addition, proteomics-based algorithms have allowed the discovery of proteins with new potential biotechnological applications from proteome data through “applied proteomics”. In this project, the first proteome analysis of the sea anemone, *Anemonia sulcata*, and its symbiont has been developed. These organisms present a wide distribution sea ecosystem. In Spain, it is accepted as a fishing and aquaculture species. Moreover, *Anemonia sulcate* has a symbiotic relation with autotroph Dinoflagellates, *Symbiodinium* spp., that allows the study of its relation at the molecular level. For the first characterization of *A. sulcata* proteome, three independent biological replicates were used, and proteins were extracted and analyzed by LC–MS/MS, allowing the quantification of 325 proteins, 81 from *Symbiodinium* spp. proteins and 244 from *A. sulcata* proteins. These proteins were subjected to gene ontology categorization by Cellular Component, Molecular Function and Biological Process. These analyzes have allowed the identification of biomarkers of gene expression as potential powerful emerging diagnostic tools to identify and characterize the molecular drivers of climate change stresses and improve monitoring techniques. In addition, through the application of novel algorithms for the detection of bioactive compounds based on the analysis of molecules of marine origin, the proteome has allowed the identification of proteins with potential applications in the fields of biomedicine and agri-food.

## 1. Introduction

The conservation and growth of natural capital accumulated in the seas and oceans are essential for the provision of sustainable ecosystem services and for the achievement of the EU’s Sustainable Development Goals (SDGs) set by the UN for 2030 [1]. Therefore, the Marine Strategy Framework Directive (MSFD) offers a comprehensive and holistic approach for the protection of the European sea, acting as an environmental pillar of the EU’s broader maritime strategy. The blue economy conveys that a healthier sea is a more productive sea. Concretely, blue biotechnology, which uses, among others, shellfish, bacteria and algae for development in health care and energy production, needs a healthy ocean, biodiversity and biomass abundance to provide innovative substances that help in the production of innovative medicines for the maintenance of human health. Moreover, marine biomass has become a potent source of new and innovative biotechnological tools for new therapeutic strategies and compounds that will increase the utility of marine biomass valorization processes and the quest for new solutions to present diseases. Therefore, the Marine offers a comprehensive and holistic approach for the protection of the European sea, acting as an environmental pillar of the EU’s broader maritime strategy [2]. The blue economy conveys that a healthier sea is a more productive sea. The potential of the oceans is enormous if we keep them healthy, hence the importance of their correct management for innovation, growth and improvement of the economy. While the economy includes existing businesses such as fisheries, coastal tourism and shipping, it also focuses on the development of new emerging sectors that were next to non-existent 20 years ago, e.g., blue carbon sequestration, marine energy and biotechnology; sectorial activities that create potential and opportunities for training and employment, but also fight climate change. Thus, the Blue Economy is linked to many other economic activities, and its impact goes beyond the sectors mentioned above [3]. Concretely, blue biotechnology, which uses, among others, shellfish, bacteria and algae for development in health care and energy production, needs a healthy ocean, biodiversity and biomass abundance to provide innovative substances that help in the production of innovative medicines for the maintenance of human health. The ability of populations to respond to increasing temperatures will play a crucial role in determining the distribution and persistence of species as global temperatures rise [4]. In the context of global climate change, it is becoming more and more demanding to anticipate alterations and responses of bioindicator species and to create a database to prevent and predict environmental and climatic changes before the damage is irreversible. Recently, the development and application of BEWS (Biological Early Warning Systems) have been reviewed using various groups of organisms (such as bacteria, algae, cladocerans, bivalves and fish) and biomarkers to process behavioral monitoring data. When proposing a candidate, a bioindicator species, in this context, it should demonstrate its importance as a blue economy supporting input, and it should have the ability to upregulate “stress-response genes” in reaction to particular stressors.

Moreover, marine biomass has become a potent source of new and innovative biotechnological tools for new therapeutic strategies and compounds that will increase the utility of marine biomass valorization processes and the quest for new solutions to present diseases. In this sense, sea anemones (order Actiniaria) are a rich source of biologically-active proteins and polypeptides. Several cytolytic toxins, neuropeptides and protease inhibitors have been identified from them. In addition to several equinatoxins, potent cytolytic proteins and an inhibitor of papain-like cysteine proteinases (equistatin) were isolated from the sea anemone *Actinia equine*. Equistatin has been shown to be a very potent inhibitor of papain and a specific inhibitor of the aspartic proteinase cathepsin D. While papain-like cysteine proteases have been implicated in various diseases of the central nervous system, such as brain tumors, Alzheimer’s disease, stroke, cerebral lesions, neurological autoimmune diseases and certain forms of epilepsy, aspartic proteinase cathepsin D is involved in the pathogenesis of breast cancer [5] and possibly Alzheimer’s disease [6]. Recently, proteomics approaches have highlighted its potential use to discover new activities with potential biotechnological applications from proteome data through “applied proteomics”. This strategy has been used to obtain a new antitumor compound from the proteome of the microalgae *Nannochloropsis gaditana* [7]. However, while a significant amount of information regarding thermo tolerance and high irradiance mechanisms has been obtained on a transcriptomic level, investigations into the proteome have so far been sparse. A narrow assortment of papers uses proteomic analysis to elaborate on fundamental cnidarian biology under stress response. In this sense, proteomic analysis techniques are considered strong tools for identifying and quantifying the protein contents in different organisms, organs and secretions. In the present research, gene expression biomarkers (GEBs) are emerging as powerful diagnostic tools for identifying and characterizing temperature and irradiance stress [8].

Among cnidarians, the extensive genomic resources now available for the non-symbiotic sea anemone *Nematostella vectensis* have opened new perspectives on the study of basal metazoans, and several EST resources have been generated for symbiotic cnidarians (predominant corals) and *Symbiodinium* spp. However, to date, only a few molecular approaches have been developed, most of them based on DNA/RNA analysis. Heat stress has been related to coral bleaching due to the loss of its autotroph symbiont. Their effects were studied using RNAseq [9], showing enriched clusters related to innate immunity, apoptosis and protein folding under heat stress conditions. Other analyses have shown that the modifications in nutrient cycling are a primary driver of the breakdown of the symbiosis [10].

In our approach, we select the sea anemone *Anemonia sulcata* as a good candidate system because it possesses the same mutualistic relationship with *Symbiodinium* spp. [11] but lacks the calcareous skeleton that hinders cellular-level work presented in other models. In addition, it is widely distributed and found in shallow temperate marine environments worldwide. Sequence characterized amplified region (SCAR) data indicate that the vast majority of *Anemonia* worldwide (encompassing two described species, *A. sucata* (Pennant, 1777) and *A. viridis* (Forsskål, 1775), appear to be genetically homogeneous. Another relevant reason to select these organisms is based on the recent discovery of molecules with antiproliferative, antioxidant, chemopreventive and antiangiogenic potential against colorectal cancer [12], including some peptides, which could be a clue to the presence of proteins with similar activities related to biomedicine or agri-food. Additionally, as has been commented, when a bioindicator species is proposed, it should demonstrate its importance as a blue economy supporting input. In Spain, *A. sulcate* is considered a fishing and aquaculture species; the impact of any improvement in its management will be directly translated into benefits for both industries. Finally, the proteome of *A. sulcate* and its symbiont *Symbodinium* spp. remains unexplored. The generation of genomic and proteomic resources would, therefore, greatly advance research addressing the understanding of symbioses at a molecular, cell-biological, and genomic level, being an optimal source of novel biotechnological tools that need to be explored. This methodology offers the capacity to collect the full list of the proteins expressed by an organism under specific conditions. While these results have been commonly used for basic approaches, recently, these approaches have been focused on the valorization of marine biomass. Most of the common therapeutic agents are proteins, with applications in agri-food and biomedicine fields, and marine biosystems possess the highest levels of diversity, being the optimal niche to search for new proteins with potential novel biotechnological applications. As an example, new proteases, antibacterial and antifungal compounds have been studied. In terms of new antitumor drugs, thousands of compounds have been isolated, around twenty are under clinical trials, and seven have been approved [13].

## 2. Results and Discussion

Protein extracts from *Anemonia sulcata* were used from protein identification to establish the first proteomic approach to this relevant marine organism in combination with its symbiotic dinoflagellate *Symbiodinium* spp. in order to develop new protein biomarkers of global warming and study the potential biotechnological applications of each identified protein.

After filtering proteomics raw data, MS/MS analysis returned a total of 615 identified proteins with quantification in at least one of the three replicates (Appendix A). Only those proteins quantified in all replicates were selected, with a total of 325 proteins remaining (Appendix A). Out of these 325, 81 were identified as *Symbiodinium* proteins (Appendix A), and 244 were identified as *Anemonia* proteins (Appendix A). These 325 proteins were selected for the core of the performed analyses. Identified proteins have been deposited to the ProteomeXchange Consortium via the PRIDE [14] partner repository with the dataset identifier PXD043202.

### 2.1. Molecular Categorization of Identified Proteins by Gene Ontology

The molecular characterization of the identified proteins from *Anemonia sulcata* and its symbiont *Symbiodinium* spp. were determined by its in silico categorization through its gene ontology annotations. To this aim, 325 identified proteins (81 of *Symbiodinium* and 244 of *Anemonia sulcata*) expressed at 17 °C and No-bleaching condition were categorized according to their specific gene ontology (GO) annotations, by Cellular component (Figure 1), Molecular Function (Figure 2) and Biological Process (Figure 3).

Cellular component categorization (Figure 1) shows three groups, common proteins and proteins belonging to the cnidarian or its symbiont. Specific proteins from *Symbiodinium* are from “Cilium”, “plastid” and “thylakoid”. On the other hand, proteins from the anemone have as origin “Endoplasmic reticulum”, “cytoplasmic vesicles”, “endosome”, “chromosome”, “nucleoplasm”, “Golgi apparatus”, “peroxisome”, “lysosome”, “extracellular space”, “vacuole”, “nucleolus”, “microtubule organizing center”, “nuclear envelope”, “nuclear chromosome”, “lipid droplet”, “extracellular region” and “extracellular matrix”. Proteins identified in both organisms belong to, among others, “cytoplasm”, “cell”, “intracellular”, etc.

Categorization of proteins according to their Molecular Function (Figure 2) was used to describe activities terms that occur at the molecular level. In our approach, “ion binding” and “oxidoreductase activity” presented the two most abundant categories in both organisms (Appendix A). The “ion binding” category included proteins related to “ADP binding”, “biotin binding”, “calcium-dependent phospholipid binding”, “GDP binding”, “manganese ion binding” and “NAD+ binding” were annotated only in the category of ion binding in *Anemonia sulcata*. Whereas “chlorophyll binding” and “iron ion binding” were annotated only in the category of ion binding in *Symbiodinium*. In “oxidoreductase activity” section, “aldehyde dehydrogenase (NAD) activity”, “3-oxoacyl-[acyl-carrier-protein] reductase (NADPH) activity”, formyltetrahydrofolate dehydrogenase activity, “glutamate dehydrogenase (NAD+) activity”, “glutamate dehydrogenase (NADP+) activity”, “glutathione-disulfide reductase activity”, “glyceraldehyde-3-phosphate dehydrogenase (NAD+) (phosphorylating) activity”, “glycerate dehydrogenase activity”, “glycerol-3-phosphate dehydrogenase [NAD+] activity”, “glyoxylate reductase (NADP) activity”, “hydroxypyruvate reductase activity”, “isocitrate dehydrogenase (NADP+) activity”, “L-iditol 2-dehydrogenase activity”, “long-chain-3-hydroxyacyl-CoA dehydrogenase activity”, “malonate-semialdehyde dehydrogenase (acetylating) activity”, “malonate-semialdehyde dehydrogenase (acetylating, NAD+) activity”, were presented only in the category of oxidoreductase activity in *Symbiodinium*. In addition, “cytoskeletal protein binding”, “enzyme binding”, “rRNA binding”, “mRNA binding”, “transcription factor binding”, “hydrolase activity, acting on carbon-nitrogen (but not peptide) bonds”, “lipid binding” and “DNA-binding transcription factor activity” were annotated only in the *Anemonia sulcata* proteome.

As a third level of protein characterization by gene onthology, the identified protein was categorized by its role in specific biological processes (Figure 3), showing that the “cellular nitrogen compound metabolic process” was the most represented category in both organisms. The second most abundant was the “biosynthetic process” in *Anemonia* and “small molecule metabolic process” in *Symbiodinium*. The detailed description included in each category is listed in Appendix A. More detailed information about all the annotated categories of Cellular Component, Molecular Function or Biological process is displayed in Appendix A.

### 2.2. Determination of Thermal Stress Biomarkers by Proteomics

Coral reefs represent one of the most relevant marine ecosystems, not only for their biological relevance but even more because of their economic relevance translated into fisheries, tourism, aesthetic importance, etc. Climate change, global warming and anthropogenic stresses have lowered the health level of these corals, producing the loss of symbiotic algae (*Symbiodinium* spp.) and its pigments. The molecular mechanism of bleaching has been widely studied using different approaches. The analysis of the whole process of coral bleaching under heat stress was studied in the sea anemone Aiptasia [9] by RNAsep. This study detected more than 500 genes involved in the process that were classified into two clusters, the first mainly related to protein folding and the second with apoptosis. In our approach, 34 proteins have been related to protein folding function (Appendix A), 6 were specifically quantified in all the replicates from *Symbiodinium* spp, 6 more were quantified in *Anemonia sulcata* and finally, 22 were quantified in all the replicates, independently of its origin. Our approach also found two proteins related to apoptosis, two common to all samples and two specifically from *A. sulcata*. To monitor the apoptotic activity under stress conditions that lead to coral bleaching, the previous report measured the caspase activity [15]. In our approach, two “caspases-3-like” proteins have been found, one common and the other from *A. sulcata*.

Those data suggest that the most common protein related to coral bleaching has been detected and quantified in our study, showing that proteomics is able to capture those protein blocks that construct complex biological scenarios, including coral bleaching. The main potential application of this approach is to develop biomarkers for coral bleaching that may be used to elucidate the level of health of our corals as a predictive tool that may be used for its management and conservation. A specific proteomic approach has been developed in our lab to capture and validate those potential global warming markers of coral bleaching.

### 2.3. A. sulcata and Symbiodinium spp. Proteomes as a New Source of Proteins with Potential Biotechnological Applications

The marine ecosystem has been considered an emerging source of new bioactive compounds [16]. However, this large amount of potential information has been ignored. Hence, it has been necessary to conduct a deep study of the physiological requirements of most marine microorganisms to enhance a greater understanding of their growth conditions [17] that promote the development of marine biotechnology, with the consequent development of novel compounds that may contribute significantly toward drug development over the next decade [16].

The potential study of marine ecosystems may elucidate new active compounds from a wide variety of organisms and microorganisms to solve problems from Alzheimer’s disease [18] to, i.e., bacterial infections [19]. On the other hand, recent developments in proteomics approaches have shown a capacity to unravel complex biological problems. An attempt to transform the large list of identified proteins from biological samples into potential biotechnological applications was developed. This methodology, named “applied proteomics”, is based on using a bioinformatic algorithm, the comparison of the list of identified proteins in a specific proteomic approach with the published list of proteins with a known bioactive role in terms of sequence, semantic, ontology, etc. Using proteomics data from the marine microalgae *N. gaditana*, a novel antitumor compound was developed, tested and patented (patent number: ES2810229A1) [7,20,21] that assumes the validation of the applied proteomics procedure.

The applied proteomics concept was developed as a tool to extract biotechnological applications from complex sets of proteome data. This system is based on a global comparison by a bioinformatic algorithm of identified proteins with the list of proteins with a confirmed bioactive role. This search is based on several aspects, such as homology, sequence, GO, etc. As a result, a text output file with all the detected connections is obtained.

Using this approximation, which has already demonstrated its usefulness with the *N. gaditana* data [22], the total 325 proteins identified in *A. sulcata* and Symbiodinium samples were subjected to applied proteomics concept to resolve those proteins with potential applications in agri-food or medicine (Figure 4). From an output text file of thousands of pages obtained from the algorithm, a search was performed to highlight potential biotechnological proteins using “cancer”, “inflammatory disease”, “Obesity and Diabetes”, “Degeneration”, “Aging”, and “antioxidant” as keywords (data not shown due to legal restrictions of patent regulations in Europe). From the total identified protein, only 14% did not present any specific activity that needs specific study, or a percentage below 0.4%. However, 68% of the identified proteins present patented activity related to cancer. In other words, most of the identified proteins present some activity related to cancer as treatments, biomarkers, or antitumors. Each of them needs to be validated by specific approaches to elucidate its potential use as a new antitumor compound. Similar conclusions may be obtained with the rest of the applied keywords, 6.1% “Inflammatory disease”, 4.9% “Obesity and Diabetes”, 4.5% “Degeneration”, 1.6% “Aging”, and 0.4% “Antioxidant”. Future research will develop the application and the biological characterization of those proteins, but in summary, the results support the initial hypothesis that these proteins will bring us new tools to develop future applications in agri-food research and biomedicine.

## 3. Material and Methods

### 3.1. Biological Samples

The specimens of *Anemonia sulcata* used in this study came from the Cantabrian Sea [23] with an in situ SST of 17 °C. Within a controlled environment chamber, each individual was randomly placed in a numbered aquarium (Figure 5) containing 5 L of natural seawater (salinity 35.5 ± 0.1 ppm) and constant aeration. Anemones were maintained at a temperature of 27 ± 0.5 °C for a 30-day period; 12 h/12 h light/dark cycle under LED tubes (Cool Daylight Philips, Madrid, Spain) and irradiance of 4 ± 0.10 quanta/cm^2^/s. *A. sulcata* was fed once per week with 3 mL of *Artemia metanauplius*, and two days after each feeding, a seawater change was performed.

### 3.2. Protein Extraction

For the characterization of the first proteome of *Anemonia sulcata*, three independent biological replicates were used. Protein extraction was performed by Fernandez-Acero et al. (2009) [24]. Obtained biomass (2 g) was lyophilized and ground into a fine powder with a mortar and pestle using liquid nitrogen. The powder was transferred to 15 mL tubes and resuspended in 10 mL of cold acetone. After vortexing for 30 s, the suspension was centrifuged (10,000× *g*, 5 min, 41 °C), and the resulting pellet was washed once again with acetone and resuspended in 10 mL 20% *w*/*v* trichloroacetic acid (TCA) in acetone. After centrifugation, the pellets were sequentially washed twice with 20% *w*/*v* TCA in acetone, once with 20% *w*/*v* TCA and finally twice with 80% *v*/*v* acetone. This pellet was air-dried and the dry powder was resuspended in 5 mL ‘‘dense SDS buffer’’ (30% *w*/*v* sucrose, 2% *w*/*v* SDS, 0.1 M Tris-HCl, pH 8.0, 5% *v*/*v* 2-mercaptoethanol). Then, 5 mL Tris buffered phenol, pH 8.0 (Biomol, Hamburg, Germany) was added, and the resulting mixture was vortexed for 30 s. The phenol phase was separated by centrifugation and transferred to a fresh tube. After addition of five volumes of cold 0.1 M ammonium acetate in methanol, the proteins were precipitated from the phenol phase overnight at 20 °C. The precipitated proteins were recovered by centrifugation, washed twice with cold 0.1 M ammonium acetate in methanol and twice with 80% *v*/*v* acetone. The final pellet was air-dried and stored at −80 °C. Obtained extracts were quantified using Qubit 2.0 Fluorometer system (Invitrogen, Carlsbad, CA, USA).

### 3.3. Protein Identification by LC–MS/MS

Protein identification was performed by Fajardo et al. [24]. In brief, protein pellets were resuspended in Laemli buffer (NuPAGE™ LDS Sample Buffer, Invitrogen, Leicestershire, UK), the extracts of proteins were cleaned up by 1D SDS-PAGE, concentrating total extract in 1 cm of resolving precast gel at 4–12% (NUPAGE, Invitrogen, Leicestershire, UK), gel slices were diced and destained with 50% of acetonitrile in 100 mM of ammonium bicarbonate. Protein digestions were carried out with trypsin in 25 mM of ammonium bicarbonate at 37 °C for 3 h. To block cysteine thiol groups, 25 mM of dithiotreitol and 40 mM of iodoacetamide in 25 mM of ammonium bicarbonate were applied to gel slices. Peptide separations in nano-LC were performed in a Dionex Ultimate 3000 nano UPLC (Thermo Fisher Scientific, Waltham, MA, USA) with a 75 μm i.d. × 50 cm Acclaim PepMap RSLC C18 column (Thermo Fisher Scientific, Waltham, MA, USA) at 300 nL/min and 40 °C. Eluting positive peptides were analyzed in positive mode on an Orbitrap Fusion™ Tribrid™ Mass Spectrometer (Thermo Fisher Scientific, Waltham, MA, USA) as described [24]. Proteomics raw data were filtered. MasterProtein information was extracted with High Protein FDR confidence (<0.01) from raw data table. In addition, contaminants and modified peptides were filtered out. Finally, quantification values were corrected by global sample intensity. Thus, each quantification value was divided by the global sample intensity and multiplied by the mean of the global sample intensities.

### 3.4. Bioinformatic Analysis

Protein extracts were loaded into an Orbitrap LC/MS (Thermo Fisher Scientific, Waltham, MA, USA). Due to the lack of molecular information in the databases of these non-model organisms, protein identification searches were performed against related organisms, the anemone *Nematostella vectensis* (UP000001593, downloaded 20 May 2021) for *Anemonia sulcata* and the symbiotic *dinoflagellate Symbiodinium microadriaticum* (*Zooxanthella microadriatica*) for its symbiont (UP000186817, downloaded 20 May 2021). Positive protein identification was accepted when the obtained values exceeded 54% of Blast Similarity Means and Blast Hits Counts > 20. From these extracts, 615 proteins were identified (Appendix A), and only those proteins quantified in all replicates were selected for in silico depth analysis. Identified proteins were annotated using OmicsBox (v.1.3.11) to verify all associated functional information. Nr databases were used as the BLASTP query search for homology, and the public EMBL-EBI InterPro was used to scan sequences against InterPro signatures. Furthermore, GO enzyme code mapping was employed to map the annotated GO terms to enzyme codes, allowing the retrieval of metabolic pathways based on the associated GO terms and enzyme codes. Finally, the GO classification of proteins identified according to Cellular Component (CC), biological process (BP) and molecular function (MF) was carried out using the Agbase web server (version 2.0) (https://agbase.arizona.edu/).

## 4. Conclusions

In the present research, we investigated the proteome of both the host, *Anemonia sulcate*, and its symbiont *Symbiodinium* spp., with the aim of gaining insight into the potential use of its proteins for specific fields such as biomedicine, aquaculture or the environment. There is an increasing need to identify novel, cosmopolitan species to be used in biomonitoring stress agents such as contamination and climate change in the coastal and open ocean. Cnidarian species have been proposed as candidates. Research studies have identified, by MS-based proteomics, proteins that have been under- or over-expressed after exposure to temperature-increase episodes. In this study, proteins involved in apoptosis and proteolysis have been quantified, with evidence of impairment to endoplasmatic reticulum and cytoskeletal regulation proteins. The results obtained regarding *Anemonia sulcata* and its symbiont, *Symbodinium* spp. (and their proteome), assures the presence of these key proteins related to the potential metabolism of defense against temperature increase as a stress agent. This is the first proteomic approach to study this cnidarian and its symbiotic dinoflagellate. New tools such as molecular biomarkers are being provided both for exposure to drivers of climate change and for the development of new biotechnological tools.

## Figures and Tables

**Figure 1 ijms-24-12798-f001:**
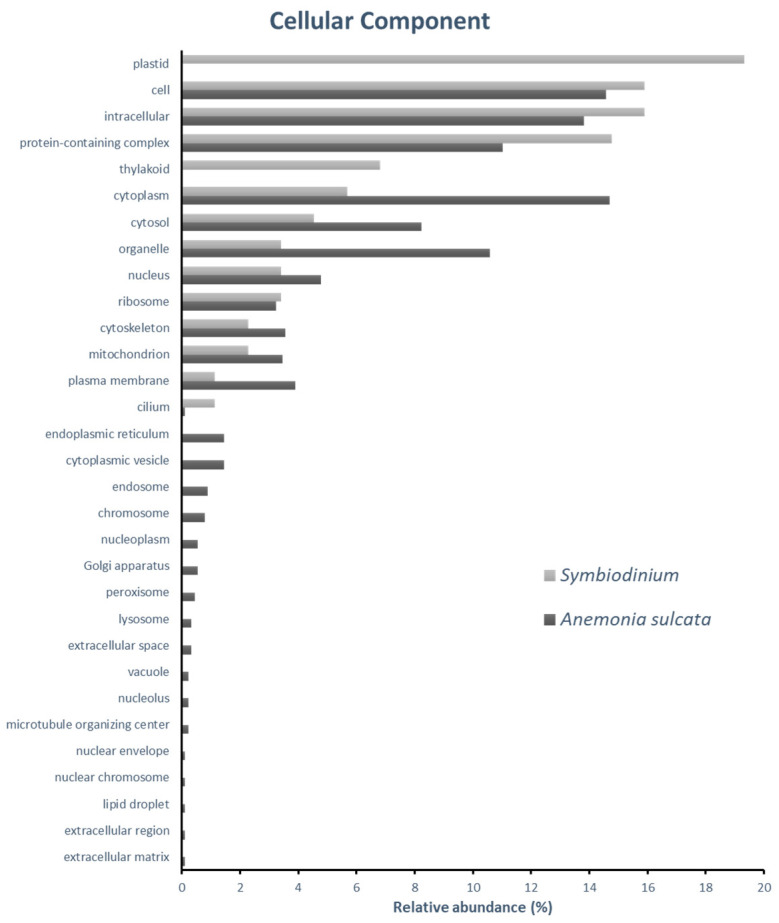
Gene ontology (GO) classification of *Anemonia sulcata* and *Symbiodinium* spp. Identified proteins by cellular component-relative abundance represent the percentage of proteins identified in each category relative to the total number of protein GO annotations at cellular component.

**Figure 2 ijms-24-12798-f002:**
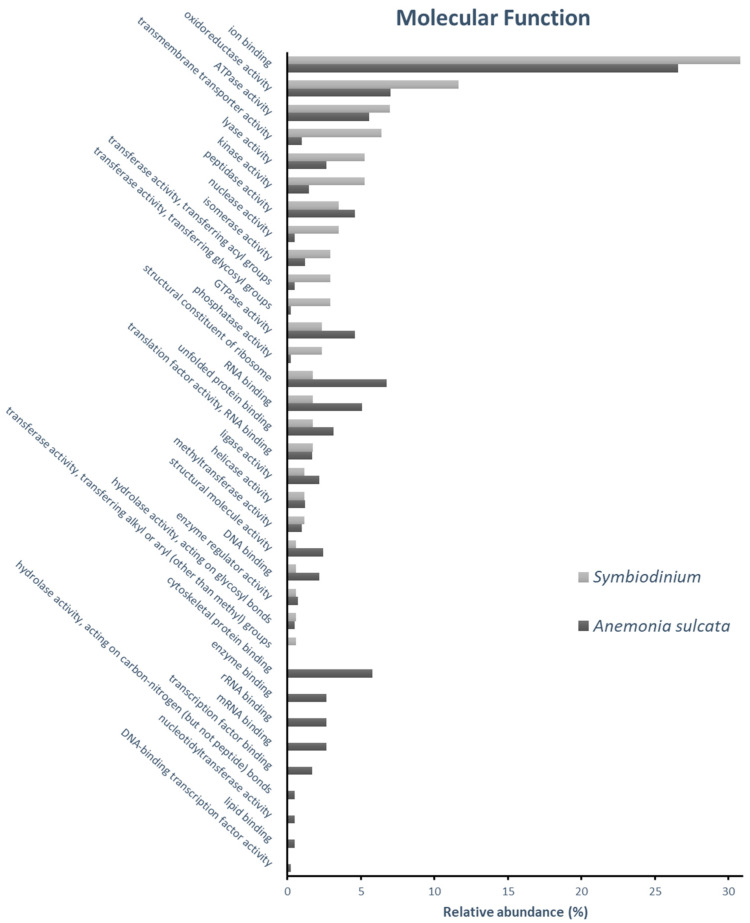
Gene ontology (GO) classification of *Anemonia sulcata* and *Symbiodinium* spp. identified proteins by molecular function-relative abundance representing the percentage of proteins identified in each category relative to the total number of protein GO annotations at molecular function.

**Figure 3 ijms-24-12798-f003:**
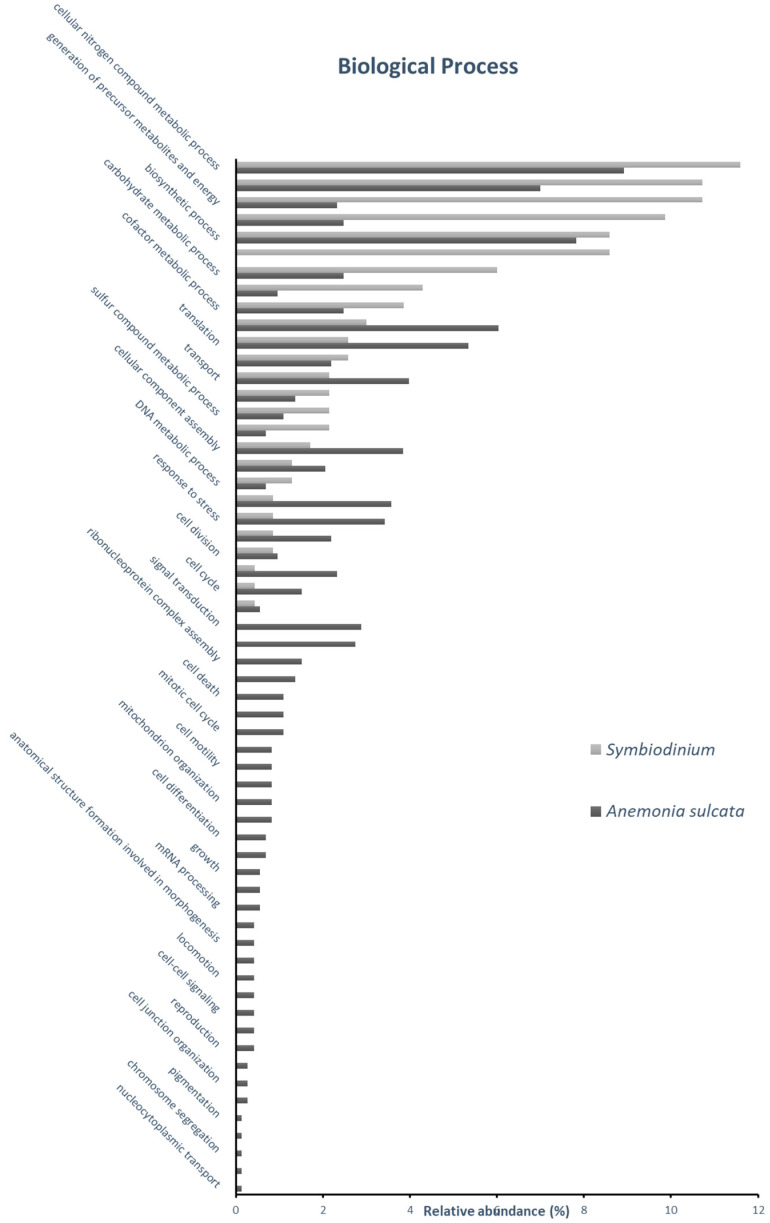
Gene ontology (GO) classification of *Anemonia sulcata* and *Symbiodinium* spp. identified proteins by biological process-relative abundance represents the percentage of proteins identified in each category relative to the total number of protein GO annotations at biological process.

**Figure 4 ijms-24-12798-f004:**
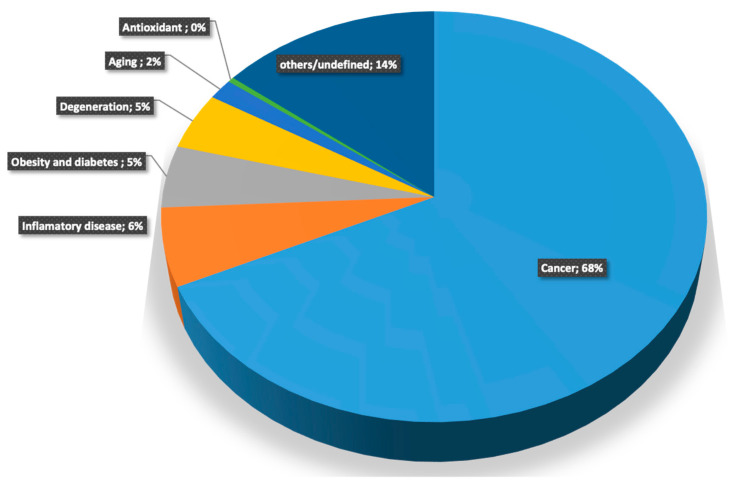
Applied proteomic algorithms results of identified proteins from *Anemonia sulcata* and *Symbiodinium* spp.

**Figure 5 ijms-24-12798-f005:**
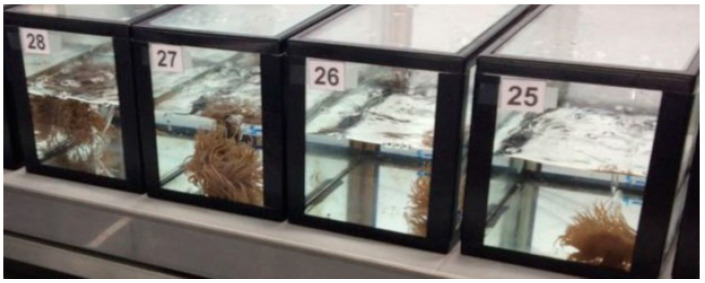
Experiment of *Anemonia sulcata* in a controlled environment chamber. One individual is in each numbered aquarium.

## Data Availability

Identified proteins have been deposited to the ProteomeXchange Consortium via the PRIDE partner repository with the dataset identifier PXD043202.

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
