# Peer review of "Proteomic Approach to Anemonia sulcata and Its Symbiont Symbiodinium spp. as New Source of Potential Biotechnological Applications and Climate Change Biomarkers"

_ijms, 2023, doi:10.3390/ijms241612798_

Round 1
Reviewer 1 Report
The work of Melendez-Perez is consistent with the scope of the journal nevertheless; it requires extensive revision before can be considered for publication.
Abstract: Assuming that this is an original research work and not a review paper, the abstract must be concise, and comprise: objectives of the study, methodology, results and conclusions. The abstract must describe very clearly the objectives of the study and what was carried out (methodological approach) and results. All redundant information must be moved to introduction…
Introduction: Likewise, much of the information in this section is redundant. I would expect to find information about the biology of the species that are being studied, and a justification for the selection of these species. Since one of the aims of the study is climate change, more specific information about this stress factor in aquatic species and cnidarian, and biomarkers already identified to monitor climate change should included (and developed) in the introduction.
Results and discussion: The authors perform a general characterization of the proteome of Anemonia sulcata along with its symbiont Symbiodinium spp, (section 2.1), and then in section 2.2 the authors identify several proteins that respond to thermal stress. Nevertheless, the authors do not explain in detail this experiment to investigate the response to temperature. Furthermore, the results (list of proteins and quantitative values) must be reported in the manuscript and not as supplementary material.
Materials and Methods: The experiment to investigate the thermal stress is not explained. Protein extraction is not described in full. How proteins are solubilized? Which buffer was used to solubilize the proteins?
The quality of English Language is in general good
Author Response
REVIEWER 1:
Comments and Suggestions for Authors
Abstract: Assuming that this is an original research work and not a review paper, the abstract must be concise, and comprise: objectives of the study, methodology, results and conclusions. The abstract must describe very clearly the objectives of the study and what was carried out (methodological approach) and results. All redundant information must be moved to introduction…
Following the reviewer suggestion, the abstract and the introduction has been rewritten to highlight the main points of our approach.
Introduction: Likewise, much of the information in this section is redundant. I would expect to find information about the biology of the species that are being studied, and a justification for the selection of these species. Since one of the aims of the study is climate change, more specific information about this stress factor in aquatic species and cnidarian, and biomarkers already identified to monitor climate change should included (and developed) in the introduction.
Following the reviewer suggestion , the introduction has been modified, deleting detected redundances, including the requested information about the selected species.
Results and discussion: The authors perform a general characterization of the proteome of Anemonia sulcata along with its symbiont Symbiodinium spp, (section 2.1), and then in section 2.2 the authors identify several proteins that respond to thermal stress. Nevertheless, the authors do not explain in detail this experiment to investigate the response to temperature. Furthermore, the results (list of proteins and quantitative values) must be reported in the manuscript and not as supplementary material.
We would like to thank to the reviewer for its comment. To avoid this misunderstanding, the main goal of our approach (the development of the first proteome analysis of the sea anemone, Anemonia sulcata and its symbiont) has been included from the abstract. The identified proteins are discussed in relation with is role in heat stress response and coral bleaching. The development of temperature experiment is running This information has been highlighted in the text Page 11 Lines 331-338). On the other hand, 2.1 we have described the Molecular categorization of identified proteins by gene ontology. 2.2 we described potential thermal stress biomarkers by proteomics and 2.3. we described potential biotechnological applications from the identified proteins.
Finally, due to the huge amount of information recovered and accordingly with common proteomics procedures and published approaches, the inclusion of a table with more than 300 lines may make this manuscript illegible. For this reason, we would like to maintain this information as excel file in supplementary material, been accessible for scientific community as a direct file for further bioinformatic developments as is usual in proteomics paper.
Materials and Methods: The experiment to investigate the thermal stress is not explained. Protein extraction is not described in full. How proteins are solubilized? Which buffer was used to solubilize the proteins?
Following previous suggestion of the reviewer, the main aim of our experiments has been included in the new version of the manuscript, where abstract and introduction has been modify accordantly. Moreover, Materials and Methods section has been modified following reviewer suggestion, including full details description of protein extraction and digestion. Proteins was cleaned using SDS Gels, for this reason after extraction, proteins pellets was solubilize in “Laemli buffer” from Invitrogen.

Reviewer 2 Report
The main focus of the article is on protein characterization using protein extracts of Anemonia sulcata, establishing the first proteomic approach to this related marine organism and its symbiotic dinoflagellate Symbiodinium spp.
Problem:
1. The articles are not innovative enough.
2. The authors suggest that cnidarian bleaching is linked to a temperature-induced alterations in protein phosphorylation.The analysis of the whole process of coral bleaching under heat stress was studied in the sea anemone Aiptasia by RNA-sep. And only the quantitative phosphatase/phosphorylase numbers are basically the same as evidence to support the practical applicability of proteomics analysis, the chain of evidence is too thin, not analyzed with a multi-omics approach, and there are insufficient examples of application.
3. The last two sentences of the article's summary are identical to the last two sentences of the introduction.
4. Fig 2, 3, and 4 are all functionally enriched diagrams, and it is recommended that they be combined into one diagram.
5. Fig 5 No figure available.
6. Suggest that the images in the article be numbered sequentially.
7. What does the underlining of lines 234-238 on page 9 mean?
Author Response
REVIEWER 2:
Comments and Suggestions for reviewer 2:
- -The articles are not innovative enough.
We would like to thanks to the reviewer for its comment. The present study is the first successfully approach to the proteome of the model system Anemona/symbiodinium. To highlight the novelty of the proposed research and according with the comments of reviewers, the abstract and the introduction has been modify accordingly.
2.-The authors suggest that cnidarian bleaching is linked to a temperature-induced alterations in protein phosphorylation. The analysis of the whole process of coral bleaching under heat stress was studied in the sea anemone Aiptasia by RNA-sep. And only the quantitative phosphatase/phosphorylase numbers are basically the same as evidence to support the practical applicability of proteomics analysis, the chain of evidence is too thin, not analyzed with a multi-omics approach, and there are insufficient examples of application.
Following the reviewer suggestion, the relation between quantitative phosphatase / phosphorylase has been deleted from the new version of the manuscript.
- The last two sentences of the article's summary are identical to the last two sentences of the introduction.
Following the reviewer suggestion, these sentences have been modify and deleted.
- Fig 2, 3, and 4 are all functionally enriched diagrams, and it is recommended that they be combined into one diagram. 5. Fig 5 No figure available. 6. Suggest that the images in the article be numbered sequentially.
Following the reviewer suggestion, figures have been modify. However, due to the Hight number of detected categories, merge figures 2,3 and 4 in one diagram will transform the new figure illegible.
Thanks to the reviewer for notifying us of the loss of figure 5 , it has been included in the new version of the manuscript. We reorganized the figures in the text, and numbered sequentially.
- What does the underlining of lines 234-238 on page 9 mean?
Thanks to the reviewer to highlighting this typographical error that has been corrected in the new version of the manuscript.

Round 2
Reviewer 2 Report
No comments.